# Chronic Pain and Work Conditions of Hotel Housekeepers: A Descriptive Study

**DOI:** 10.3390/ijerph19063383

**Published:** 2022-03-13

**Authors:** Cristian Sánchez-Rodríguez, Oana Bulilete, Xenia Chela-Alvarez, Olga Velasco-Roldán, Joan Llobera-Canaves

**Affiliations:** 1Primary Care Research Unit of Mallorca, Public Health Service of the Balearic Islands (Ibsalut), 07003 Palma, Spain; oana.bulilete@ssib.es (O.B.); xenia.chela@ibsalut.es (X.C.-A.); jllobera@ibsalut.es (J.L.-C.); 2Research Group on Global Health and Human Development, University of the Balearic Islands (UIB), 07122 Palma, Spain; 3Research Group in Primary Care and Promotion—Balearic Islands Community (GRAPP-caIB), Network for Research on Chronicity, Primary Care, and Health Promotion (RICAPPS), 07003 Palma, Spain; 4Health Research Institute of the Balearic Islands (IdISBa), 07120 Palma, Spain; olga.velasco@uib.es; 5Research Institute of Health Sciences (IUNICS), Department of Nursing and Physiotherapy, University of the Balearic Islands (UIB), 07122 Palma, Spain

**Keywords:** occupational health, musculoskeletal pain, occupational risk, perceived health status, hotel maids

## Abstract

Objectives: To estimate the prevalence of musculoskeletal pain of hotel housekeepers (HHs) and to describe the work conditions and perception of health in this occupational group in the Balearic Islands, Spain. Methods: Cross-sectional descriptive study with HHs of the Balearic Islands, performed in primary care. Random sample of HHs who worked during the 2018 season. We collected information on sociodemographic variables, job characteristics, workload, pain, perceived health, and physical activity. After participants signed the informed consent form, we conducted a face-to-face interview in the primary care centre and accessed the participants’ electronic health records. Results: 1043 HHs aged 43.3 ± 10 years and with 10.7 ± 9.1 years worked as HHs were included. 51% (95% CI: 48–54%) reported chronic pain, mainly in the lower back 28.7% (95% CI: 25.9–31.5%), hands/wrists 23.7% (95% CI: 21.1–26.4%), neck 21.6% (95% CI: 19.1–24.3%), shoulders 19.9% (95% CI: 17.4–22.4%), and back 17.8% (95% CI: 15.4–20.2%). Pain was associated with older age, more years worked, more beds made/day and difficulty in pushing the housekeeping cart. More than half HHs reported that they did not incorporate occupational risk prevention measures (ORPMs) into their routine; 17.3% (95% CI: 15.1–19.7%) HHs considered their health as poor or very poor. Perception of health was worse in HHs with chronic pain. Conclusions: A high percentage of HHs of the Balearic Islands reported chronic pain, a low compliance with ORPMs and compared to women of the same sociodemographic profile they perceive a worse health status.

## 1. Introduction

According to the World Health Organization, musculoskeletal pain is the main occupational disease [1]. Within the hospitality sector, hotel housekeepers (HHs) have the highest rate of musculoskeletal disorders (MSD) and acute traumatic events [2]. The main complaint of this population is musculoskeletal pain, particularly in the neck, lower back, shoulders, hands, wrists, and knees [3,4,5].

It has been suggested that HHs are at high risk for MSD [5], due to age [6], sex [2] and organisational characteristics. Currently, the horizontal and vertical segregation of the marketplace concentrates women in jobs with high time pressure and heavy workload [7,8].

Occupational risk factors can be physical, chemical, biological, and psychosocial [9]. Most studies evaluate physical risks and harmful outcomes derived from long-term exposure. Physical risk factors include repetitive work, use of excessive force when lifting/moving weights, awkward postures, manual loading of objects, standing during long periods, elevation of the upper limbs and insufficient breaks [4,5,7,10,11].

Assessment of the physical burden of HHs is complex, since it varies in each workplace based on the number and type of rooms/day, the number of checkout rooms/day, the number and type of beds/day, and other aspects. While there is no standardized instrument to measure all dimensions of physical risks in HHs [12], it is crucial to evaluate them because MSDs are the most common occupational conditions [13], they frequently become chronic [14] and they are preventable according to the European Agency for Safety and Health at Work [15].

Tourism is the main economic sector of the Balearic Islands. In 2018, it constituted 44% of the annual GDP of the region [16]. A total of 80,461 people (14.02% of employed workers) were employed in the hospitality sector [17], with HHs comprising the largest group.

The framework for this study is the project *Hotel Housekeepers and Health* (ITS’17-096). This project follows the structured phases proposed by the British Medical Council (BMC) regarding trials for complex interventions [18] to promote changes in behaviours, improve feasibility and facilitate translation of effective outcomes into practice. This study is part of Phase 1, with the secondary objective of modelling a future intervention.

The main objective of this study is to estimate the prevalence of musculoskeletal pain in HHs of the Balearic Islands and to describe their sociodemographic characteristics, work conditions and perception of health.

## 2. Materials and Methods

A descriptive cross-sectional study was conducted between November 2018 and February 2019 in 39 of the 58 primary healthcare centres (PHCs) of the Balearic Islands. Participants were contacted by telephone and asked for inclusion criteria, and set up a date for the interview, which was carried out in the Primary Care Health Centre, where each HH had her general practitioner. We conducted face-to-face interviews during the winter months, because most HHs were not working and thus more available.

The study population was comprised of HHs who had worked during the 2018 summer season and had a public health services card for the Balearic Islands. The 39 centres with the highest proportion of HHs were selected. A simple random sampling of HHs attending these 39 PHCs was carried out. Informed consent was obtained from all subjects involved in the study.

Exclusion criteria included: HHs under 18 years of age; HHs unable to go to the PHC; HHs who did not have a telephone; and HHs with a language barrier.

Information of HHs was provided by the Balearic Occupational Service to the Balearic Public Health Service. This information allowed us to identify to which primary care health centre they belong and their contact data. We obtained 13,000 potential eligible participants. Initially, we calculated that a sample size of 987 HHs was needed, with a 3% precision, 95% confidence and a maximum indeterminacy (*p* = 1 − *p*). In our study, the prevalence of chronic pain was 51%; then, we can estimate this prevalence with a 2.9% precision. Estimating a 30% drop-out rate, we selected 3 additional HHs of similar age of the same PHC per participant.

### 2.1. Used Methods and Scales

Sociodemographic variables. Age, country of origin, nationality, and educational level were collected.

Occupational risk variables. We included type of contract, place of work, years worked as HH and workload (months worked in the previous year, number of rooms cleaned/day and number of beds made/day). In addition, we collected information on the use and maintenance of cleaning equipment and knowledge about occupational risk prevention measures (ORPMs), as stated in the Manual for the Prevention of Ergonomic and Psychosocial Risks of the National Institute for Occupational Safety and Hygiene [19].

Variables related to pain. We used the Nordic Standardized Questionnaire for Musculoskeletal Symptoms in the working population [20]. We asked about the frequency of pain in the previous season, with four response options (“never”, “sometimes”, “often”, and “chronic”), and in the 7 days prior to the interview (yes/no answer). Consumption of painkillers and use of alternative or complementary medication as reported during the interview were registered. Additionally, data on the medication prescribed at two time points (July and December 2018) were collected.

Physical activity (PA) was collected using the Spanish version of the Brief Physical Activity Assessment Tool [21]. This questionnaire assesses moderate and intense PA; moderate PA uses a scale of 0–4, while intense PA ranges from 0 to 2 points. When a person meets PA recommendations, the sum of the two PA is 4 points or more.

Perceived health during the previous year was recorded on a Likert-5 scale from “very bad” to “very good” according to the National Health Survey [22], and on a visual scale of 0–100 points on the day of the interview. We also calculated and recorded the BMI.

Finally, we collected willingness to participate in prevention activities during the months with less workload.

### 2.2. Data Collection

The HHs were contacted by telephone and invited to participate in the study. If they consented to participate, an appointment was scheduled at their own PHC to conduct an individual interview lasting approximately one hour. Interviews were conducted by previously trained staff with extensive experience in health research studies. The pharmacological treatment was later verified in the electronic health records of the HHs.

### 2.3. Statistical Analysis

We provide absolute and relative frequencies and 95% confidence intervals (95% CI) for categorical variables, and mean, standard deviation (SD) and 95% CI for numerical variables. We used the Chi-square test (χ^2^) for contrasts between qualitative variables. Statistical significance was set at 0.05. ORs were calculated for the personal variables and job characteristics in relation to the chronic pain variable. We used binary logistic regression with the backward method for the independent variables with a significance level *p* < 0.05. Missing data were treated as not valid. We used SPSS software (version 23, IBM, Chicago, IL, USA) for analysis.

## 3. Results

### 3.1. Participants

The study included a total of 1043 HHs: 773 in Mallorca, 137 in Ibiza, 89 in Menorca and 44 in Formentera. Figure 1 details the recruitment process.

### 3.2. Personal and Workplace Characteristics

Table 1 shows the sociodemographic and occupational variables. The participants were mainly Spanish (54%), had a mean age of 43.3 ± 10 years and had been working as HHs for 10.7 ± 9.1 years.

### 3.3. Musculoskeletal Pain

A total of 51% (95% CI 47.97–54.03%) HHs explained having suffered from chronic pain during the previous season in at least one body region. In contrast, 7.6% (95% CI 6–9.2%) declared that they just had felt pain “sometimes” (6.4%) or “never” (1.2%) during the previous season. The five body regions most affected by chronic pain were the low back (28.8%), wrists/hands (23.7%), neck (21.7%), shoulders (19.7%) and back (17.8%). Table A1 in Appendix A shows prevalence of pain by body region and age group.

A positive correlation (*p* < 0.001) was found between years worked as HH and proportion of chronic pain. Table 2 shows the distribution of chronic pain globally and in the five most affected body regions by years worked as HH. Notably, prevalence of chronic low back pain in HHs who have worked >20 years is 45% (95% CI 36.3–53.9%).

Older age (*p* < 0.001), greater number of beds/day (*p* < 0.001), more years worked as HH (*p* < 0.001), type of contract (*p* < 0.05) and greater number of hours worked per week (*p* < 0.05) positively correlated with chronic pain. Similarly, chronic pain was associated with a worse perception of health (*p* < 0.001). In contrast, physical activity and BMI were not statistically associated with pain during the previous season. Table 3 shows the relation between chronic pain and sociodemographic and labour variables. Additionally, Table 4 shows the chronic pain explanatory model with the ORs of the most relevant work and personal variables. The logistic regression model was performed with the initially statistically significant variables.

With respect to pain during the previous week, 79.2% (95% CI 76.6–81.5%) HHs reported having felt pain in at least one body region. Notably, 92.6% HHs who suffered from chronic pain in at least one body region during the previous season also expressed having felt pain during the week prior to the interview. Figure 2 shows pain frequencies for each body region during the previous week.

### 3.4. Use of Medicines

Up to 54.7% (95% CI 51.7–57.7%) of HHs explained that they had taken painkillers during the previous year and 17.7% (95% CI 15.5–20.2%) had taken alternative or complementary medication.

### 3.5. Occupational Risk Factors and Workload

Although 83.6% HHs reported that they were aware of occupational risk prevention measures (ORPMs) and 83.3% stated that they “always” attended the training courses, only 61.5% considered the information and training received as “always” suitable for the job. Additionally, 81% explained that they never stretched or warmed up before work. Regarding ORPMs, HHs reported that they were “never implemented”: by 55.5% HHs when awkward postures were required (e.g., cleaning the bathtub, making beds); by 55.7% HHs when performing repetitive movements (e.g., scrubbing, vacuuming); and by 52% HHs when handling loads (e.g., pushing the cart, carrying weights > 5 kg). Moreover, 53.3% stated that they did not have the assistance of a male room attendant/brace to lift heavy loads. In addition, 41.5% answered that taking breaks was possible “only once” (14.1%) or “never” (27.4%), whereas 33.7% reported that they could “always” take a break.

Regarding workload, HHs had worked an average of 7 months during the 2018 season, cleaning an average of 18 (±6.5) stay-over rooms/day, 5.4 (±2.8) checkout rooms/day and making 44.6 (±20.7) beds/day.

Finally, when asked to assess the adequacy of the equipment, 59.9% and 50.5% considered the housekeeping cart too heavy and too cumbersome, respectively.

Following ORPMs was significantly associated (*p* < 0.05) with a reduction of chronic pain in the spine and in the wrists/hands when handling loads and performing repetitive movements.

### 3.6. Physical Activity, BMI and Perceived Health Status

Table 5 shows health characteristics. A majority of HHs (57%, *n* = 1040) were considered insufficiently active, since they did not reach the minimum recommended PA. Up to 27.9% and 80% stated that they never performed moderate (0 points, *n* = 1041) or intense PA, respectively.

The BMI mean was 25.7 (SD = 4.9) and 49.1% (95% CI 45.9–52.2%) of the HHs were overweight or obese.

The average perceived health status on the day of the interview was 72.4 points (SD ± 18.9). Regarding the perceived health status in the previous year, 58.9% (95% CI 55.8–61.9%) evaluated their health as “poor” (“fair”, “bad” and “very bad”).

Finally, and with the objective of designing a future intervention, HHs were asked if they would attend sessions with the goal to improve their quality of life, and where they would like to have these sessions. A total of 90.3% (*n* = 1037) answered that they would be willing to attend these sessions, and 70% of the respondents (*n* = 932) would choose to have them in their PHC. Only 3.4% of HHs chose their workplace for an intervention aimed at improving their quality of life.

## 4. Discussion

To the best of our knowledge, this is the largest descriptive study conducted in HHs. Additionally, the study has been conducted in a region where tourism is the main economic sector. This random sample represents all HHs and hotel types of the Balearic Islands.

We should underscore the high prevalence of chronic pain, which affects mainly, in descending order, the low back, hands/wrists, neck, shoulders and back. Regarding occupational risk factors, over 50% participants stated that they did not follow ORPMs. The workload of the HHs was also estimated.

There is a lack of studies that evaluate pain and MSDs in HHs, especially in Spain and Europe. However, international results are comparable to ours. A survey from the five biggest national companies in the U.S. found a 91% prevalence of pain, mainly affecting the spine and upper limbs [23]. A study carried out in Las Vegas estimated 75% prevalence of pain during the previous 12 months [24]. More recently, studies in Ethiopia observed that 58.1% of HHs reported pain in the low back [25] and 62.8% in the neck and upper limbs [26]. The lower prevalence of pain in the Ethiopian studies could be explained by the much younger average age.

Importantly, age is one of the statistically significant variables regarding pain. It has been suggested that the relationship between age and pain could be explained by the accumulation of exposure [27], also called allostatic load [28], and age-related degeneration of joints and loss of muscle strength [26]. In the explanatory model of chronic pain shown in Table 4, HHs over 55 years have an OR = 4.16 of suffering from chronic pain compared to HHs under 35 years of age.

The low back is the most studied body region in HHs. However, data collection regarding prevalence of pain can refer to whether pain is present or absent and intensity or frequency of pain, which might explain discrepant prevalences among studies [5,23,25]. In this work, the low back was the most affected region both in the previous 7 days and chronically. Almost half of the interviewed HHs explained that they had had low back pain in the previous 7 days, despite most of them not actively working at the time. This would also illustrate the severity of the pain, which persists in the absence of work exposure factors.

Our data reveal that over half of HHs claimed to have suffered from chronic pain in at least one body region during the previous season. The highest prevalence of pain was found in the low back (28.8%) and the neck (21.7%). In women of the same sociodemographic profile (sociodemographic status VI-unskilled workers), the Spanish National Health Survey estimates prevalence of chronic pain of 26.7% and 23.3% in the low back and neck [22], respectively. Nevertheless, this survey refers to the presence of chronic pain in a longer period of time, the previous 12 months.

We believe that the differences in the prevalence of pain compared to other studies are due to the sociodemographic characteristics of the population and not to improvements in ergonomics or workload, which have been singled out for targeted intervention in various studies [26,29,30].

Regarding the protective effect of PA, 57% of HHs in our study were considered insufficiently active. In contrast, the proportion of HHs who did not perform PA reached 94.1% in other studies [26]. We believe that interviewing out of the work season might account for this difference, because HHs have more time to exercise. However, because they had started exercising just a few weeks before the interviews, the beneficial effects of PA may not have manifested yet, which could explain the lack of association between PA and pain.

With regard to ORPMs, results indicate that most HHs consider that the information provided in the training sessions, which most of them attend, is not transferable to their work routine. We hypothesise that the high workload prevents the implementation of the recommendations when adopting awkward postures. However, future research needs to address the exact reasons behind these behaviours. The most important exposure factors associated with pain are reaching out, overstretching and repetitive movements [23,25,26].

It is estimated that, while cleaning a room, HHs might change posture every 3 s, which amounts to approximately 8000 posture changes per shift [31]. The high frequency of posture changes and scarce compliance with ORPMs could underlie the pathophysiology of pain in HHs. Our results indicate a chronic pain OR = 2.56 for the HHs who did not follow ORPMs.

Up to 53.3% of HHs reported that no male room attendant/braces were available to assist in lifting heavy loads. It should be noted that improving ergonomics during lifting can prevent MSD [30]. Interestingly, 32% of this 53.3% acknowledged suffering from chronic low back pain. In agreement with other authors, most HHs explained that the housekeeping cart is too heavy and cumbersome, and thus inadequate for their job [32]. Improving ergonomics remains crucial for the occupational health of HHs (9,32).

Work breaks can also reduce the harmful effects of the HHs’ duties (6,30). In our study, the OR of chronic pain was 1.56 in participants who could not take any breaks. We emphasize that incorporating breaks during work hours has proven effective and should be integrated in the ORPMs of all hotel businesses [9].

Regarding perceived health status, 17.3% evaluated their health as “bad” or “very bad”, and 41.6% as “fair”. In contrast, the Spanish National Health Survey of the same year for women with the same sociodemographic profile shows a perceived “good” health status in 44.95%, and a perceived poor health status in a lower 10.88%, with “bad” and “very bad” in 7.87% and 3.01%, respectively [22]. Regarding the visual health status tool, HHs reported a mean score of 72.4 (±18.9) points out of 100. In contrast, the mean score for women with the same sociodemographic profile was 86.65 (±11.37) points [22]. The HHs of the Balearic Islands have a worse perception of their own health than the average women of the same sociodemographic profile. This discrepancy might respond to work conditions under temporary contracts, the imbalance between effort invested and rewards, a low decision margin and a high workload [32,33,34].

The prevalence of pain in the HHs of the Balearic Islands agreed with other studies and settings. We believe that interventions aimed to raise the quality of life, reduce pain, and improve the perceived health status of HHs should be urgently implemented. Solutions should promote physical activity, reduce stress, improve work–life balance, decrease workload and encourage the uptake of ORPMs [29,33,35]. Interestingly, PHCs emerged as the HHs’ favourite setting to conduct health interventions.

## 5. Limitations

This study has some limitations. We should consider the possibility of memory bias, since self-reports were used. Although we administrated a validated pain questionnaire, no criteria for chronicity were explained, and the severity of pain was completely based on subjective criteria [18]. The interviewers did not have health degrees but had been adequately trained and supervised. Since the interviews were conducted outside the tourist season and only one in six HHs had worked during the week prior to the interview, the immediate effects of the job are possibly understated.

## 6. Conclusions

HHs of the Balearic Islands present a high prevalence of chronic pain and a low compliance with ORPMs. Compared to women of the same sociodemographic profile, they perceive a worse health status; therefore, interventions to improve the health of HHs are urgently needed.

## Figures and Tables

**Figure 1 ijerph-19-03383-f001:**
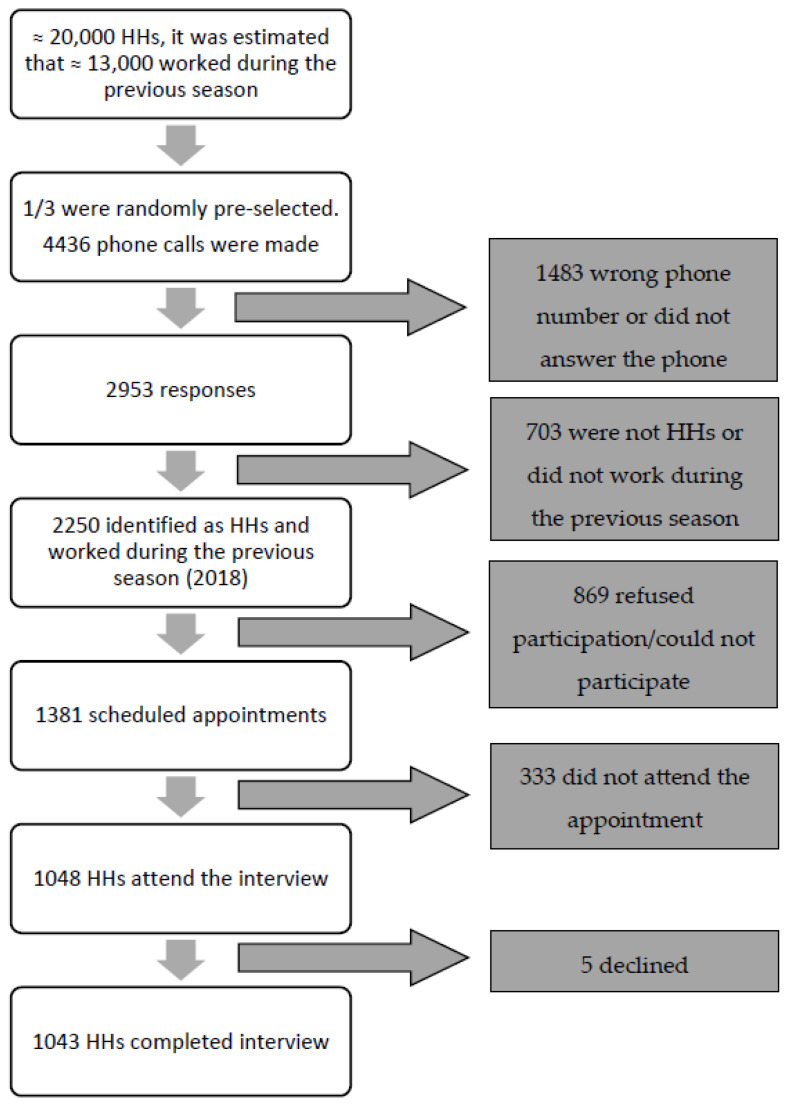
Recruitment flowchart.

**Figure 2 ijerph-19-03383-f002:**
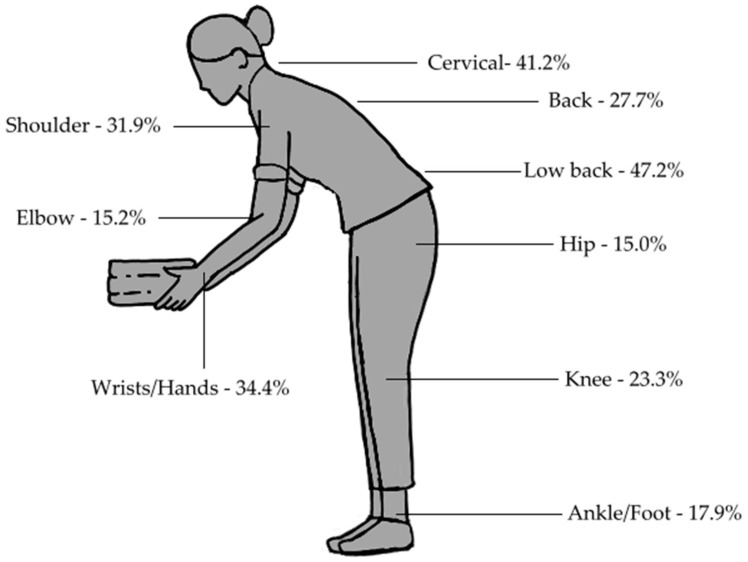
Pain by body region in the previous 7 days.

**Table 1 ijerph-19-03383-t001:** Sociodemographic and occupational characteristics.

Variables		Total Sample (*n* = 1043)	*n*	%
Age in years (*n* = 1036)		Mean (SD)	43.3 (10.1)	
	Groups	≤35	217	20.9
		35 to 44	354	34.1
		45 to 54	293	28.3
		≥55	173	16.7
Nationality (*n* = 1043)		Spanish	563	54.0
		Dual nationality	182	17.4
		Foreign	298	28.6
Educational level (*n* = 1041)		Unfinished primary	34	3.3
		Primary education completed	177	17.0
		Secondary education (first cycle)	414	39.8
		Secondary education (second cycle)	356	34.2
		University degree	60	5.8
**Workplace characteristics**				
Years worked as HH (*n* = 1019)		Mean (SD)	10.7 (9.1)	
	Groups	<10 years	562	55.2
		10–20 years	325	31.9
		>20 years	132	13.0
Type of contract (*n* = 1035)		Permanent	63	6.2
		Recurring fixed-term	551	54.2
		Temporary	402	39.6
Workplace (*n* = 1116) *		Hotel	689	66.0
		Apartment (tourist apartment, hotel apartment…)	358	34.4
		Other (agritourism, bed and breakfast, rural accommodation…)	66	6.3
Star rating (stars/keys) (*n* = 1025) **		None, 1, 2 or 3 stars/keys	283/67	29.5/89.3
		4 stars/4 keys	574/5	59.8/6.7
		5 stars/4 keys superior	103/3	10.7/4.0
Hours of work/week (*n* = 1042)		Mean (SD)	40.7 (5.6)	
	Groups	<40 h/week	68	6.5
		40 h/week	756	72.5
		41–50 h/week	196	18.8
		>50 h/week	22	2.1

* 73 HHs worked in more than one type of business. ** 10 HHs worked in more than one type of business.

**Table 2 ijerph-19-03383-t002:** Bivariate analysis of chronic pain by body regions and years worked (*n* = 1019).

		Total		
Years Worked	Region of Chronic Pain	*n*	%	95% CI
**<10 years worked**	Low back	130	23.9	20.4–27.7
	Wrists/Hands	98	18.1	1.9–21.6
	Neck	96	17.6	14.5–21.1
	Shoulders	86	15.9	12.9–19.4
	Back	85	18.9	12.9–19.2
**10–20 years worked**	Low back	96	29.9	24.9–35.2
	Wrists/Hands	86	27	22.2–32.3
	Neck	76	23.7	19.1–28.7
	Shoulders	70	21.8	17.4–26.7
	Back	55	17.2	13.3–21.8
**>20 years worked**	Low back	59	45	36.3–53.9
	Wrists/Hands	50	38.2	29.8–47.1
	Neck	44	33.8	25.8–42.7
	Shoulders	40	30.8	22.9–39.5
	Back	36	27.5	20–35.9

**Table 3 ijerph-19-03383-t003:** Relation between chronic pain and sociodemographic and labour variables.

Variables	Groups	Chronic Pain *n* (%)	*p*-Value
Age	<35 years	77 (35.6)	0.000
	35–44 years	175 (49.4)	
	45–54 years	157 (53.6)	
	≥55 years	119 (68.8)	
Years worked as HH	0–4 years	117 (36.3)	0.000
	5–9 years	130 (54.2)	
	10–14 years	87 (50.3)	
	15–19 years	66 (59.5)	
	≥20 years	120 (69.4)	
Number of beds/day	0–30 beds/day	97 (36.9)	0.000
	31–44 beds/day	137 (48.6)	
	45–60 beds/day	163 (56.6)	
	>61 beds/day	135 (65.9)	
Type of contract	Permanent	26 (41.3)	0.002
	Recurring fixed term	308 (55.9)	
	Temporary	184 (45.8)	
Number of hours worked/week	<40 h/week	25 (36.8)	0.025
	40 h/week	403 (53.3)	
	41–50 h/week	91 (46.4)	
	>50 h/week	13 (59.1)	
Perception of health	Very good	17 (27)	0.000
	Good	132 (36.2)	
	Fair	248 (57.3)	
	Bad	91 (74.6)	
	Very bad	42 (72.4)	
Physical activity	Insufficiently active	307 (51.8)	0.548
	Sufficiently active	223 (49.9)	
BMI	Underweight	10 (47.6)	0.074
	Normal weight	245 (49.9)	
	Overweight	182 (55.3)	
	Obese	80 (49.38)	

**Table 4 ijerph-19-03383-t004:** Explanatory model of chronic pain. Binary logistic regression.

	*p*-Value	OR	95% CI
Age			
<35 years (ref.)			
35–44 years	0.08	1.67	1.14–2.44
45–54 years	0.00	2.51	1.45–3.20
>55 years	0.00	4.16	2.59–6.66
Number of beds			
<30 beds (ref.)			
31–44 beds	0.66	1.09	0.74–1.59
45–60 beds	0.07	1.68	1.15–2.46
>60 beds	0.00	2.44	1.60–3.74
Scheduled breaks			
Always (ref.)			
Very often	0.03	*0.55*	0.32–0.94
Sometimes	0.36	1.24	0.77–2.01
Seldom	0.11	1.50	0.90–2.49
Never	0.00	1.56	1.11–2.18
Switches tasks with colleagues			
Always (ref.)			
Very often	0.41	1.25	0.72–2.17
Sometimes	0.58	1.14	0.70–1.86
Seldom	0.20	1.43	0.81–2.50
Never	0.00	1.80	1.18–2.74
Implements protocol of prevention of occupational risk when handling heavy loads			
Always (ref.)			
Very often	0.15	1.56	0.84–2.86
Sometimes	0.14	1.54	0.86–2.75
Seldom	0.00	2.18	1.21–3.93
Never	0.00	2.36	1.44–3.84

**Table 5 ijerph-19-03383-t005:** Health characteristics.

			*n*	%	95% CI
Physical activity (*n* = 1040)	Mean (SD)	2.6 (2.3)			
	Groups	Insufficiently active	593	57.0	54.0–60.0
		Sufficiently active	447	43.0	40.0–46.0
BMI (*n* = 1003)	Mean (SD)	25.7 (4.9)			
	Groups	Underweight (<18.5)	21	2.1	1.4–3.2
		Normal weight (18.5–24.9)	491	49.0	45.9–52.0
		Overweight (25–29.9)	329	32.8	30.0–35.8
		Obese (>30)	162	16.2	14.0–15.6
Perceived health status (*n* = 1041)	Mean (SD)	72.4 (18.9)			
	Groups	Very good	63	6.1	4.8–7.7
		Good	365	35.1	32.2–38.0
		Fair	433	41.6	38.6–44.6
		Bad	122	11.7	9.9–11.8
		Very bad	58	5.6	4.3–7.1

## Data Availability

The data presented in this study are available upon reasonable request from the corresponding author.

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
