# Peer review of "Chronic Pain and Work Conditions of Hotel Housekeepers: A Descriptive Study"

_ijerph, 2022, doi:10.3390/ijerph19063383_

Round 1

Reviewer 1 Report

This is an interesting and important study on the prevalence of MSD in hotel housekeepers. I would recommend to add the following information:

  1. Methods: I did not fully understand the recruiting procedure. The authors called the HHs and they made an appointment in the health clinic for the interview? Where did they get the list of HHs from? I would recommend to add a small paragraph concerning the recruitment of the participants.
  2. Methods: What was the endpoint and the assumption for the sample size calculation?
  3. Results: Information on BMI, physical activity and overall perceived health status should be added in Table 1 or an additional table regarding health characteristics.
  4. Results: I would also recommend an additional table regarding the information beginning with L151: the correlation analysis between pain and important confoundign factors, such as age and BMI should be seperately shown. This was very interesting to read and needs more attention.

Overall, this is a nice study.

Reviewer 2 Report

Thank you for the opportunity to review the paper: Chronic Pain and Work Conditions of Hotel Housekeepers: Descriptive Study.

Paper is interesting and important from the view of practice.

Here are some remarks:

I would be more specific about giving data in abstract on chronic pain, mainly in the low back, hands / wrists, neck, shoulders, and back – these actually are most of WrMSD, data on ratio between body parts would be more interesting.

In the part of - 2.1. Information, there is text: “Sociodemographic variables. Age, country of origin, nationality, and educational level. “ it doesn’t have  structure of sentence and paper. I wuld also change the subchapter name: “2.1. Information,” – it s Used method and scales, or at least information on methods used

Also “2.2. Study development” is confusing – more data collection?

Figure 1. Recruitment flowchart. should be in 2. Materials and Methods – it would aldso fix structure of chapter 3, where there is 3.1. and this meantime figure in the part without subchapter

Table 1. Sociodemographic and occupational characteristics should be fixed – it is difficult to read – it can be narrowed by eliminating Frequency (n) and instead n (actually its number not frequency) instead of  Percentage (%) – use “%” Here I would always use the same number of digits – not 17, but 17,0

I like the idea of Figure 2. Pain by body region in the previous 7 days, but there is always pain  - it could be even more graphical elaborated.

I think that Table 3. Explanatory model of chronic pain. Binary linear regression. could be elaborated better – having this data can be used for a purpose of structural modeling.

Nerveless the paper has its potential and should be highly citable. 
